# Pushing the Limits of Few-shot Anomaly Detection in Industry Vision: GraphCore

**Guoyang Xie**[1,2,*] **Jingbao Wang**[1,*,†]**, Jiaqi Liu**[1,*]**, Feng Zheng**[1,†]**, Yaochu Jin**[2,3]

[1]Research Institute of Trustworthy Autonomous Systems,
Southern University of Science and Technology, Shenzhen 518055, China
[2]NICE Group, University of Surrey, Guildford, GU2 7YX, United Kingdom
[3]NICE Group, Bielefeld University, Bielefeld 33619, Germany
`guoyang.xie@surrey.ac.uk`, `linkingring@163.com`
`liujq32021@mail.sustech.edu.cn`, `zhengf@sustech.edu.cn`
`yaochu.jin@uni-bielefeld.de`

## Abstract

In the area of few-shot anomaly detection (FSAD), efficient visual feature plays an essential role in the memory bank $\mathcal{M}$-based methods. However, these methods do not account for the relationship between the visual feature and its rotated visual feature, drastically limiting the anomaly detection performance. To push the limits, we reveal that rotation-invariant feature property has a significant impact on industrial-based FSAD. Specifically, we utilize graph representation in FSAD and provide a novel visual isometric invariant feature (VIIF) as an anomaly measurement feature. As a result, VIIF can robustly improve the anomaly discriminating ability and can further reduce the size of redundant features stored in $\mathcal{M}$ by a large amount. Besides, we provide a novel model GraphCore via VIIFs that can fast implement unsupervised FSAD training and improve the performance of anomaly detection. A comprehensive evaluation is provided for comparing GraphCore and other SOTA anomaly detection models under our proposed few-shot anomaly detection setting, which shows GraphCore can increase average AUC by 5.8%, 4.1%, 3.4%, and 1.6% on MVTec AD and by 25.5%, 22.0%, 16.9%, and 14.1% on MPDD for 1, 2, 4, and 8-shot cases, respectively.

## 1 Introduction

With the rapid development of deep vision detection technology in artificial intelligence, detecting anomalies/defects on the surface of industrial products has received unprecedented attention. Changeover in manufacturing refers to converting a line or machine from processing one product to another. Since the equipment has not been completely fine-tuned after the start of the production line, changeover frequently results in unsatisfactory anomaly detection (AD) performance.

How to achieve rapid training of industrial product models in the changeover scenario while assuring accurate anomaly detection is a critical issue in the actual production process. The current state of AD in the industry is as follows: (1) In terms of detection accuracy, the performance of state-of-the-art (SOTA) AD models degrades dramatically during the changeover. Current mainstream work utilizes a considerable amount of training data as input to train the model, as shown in Fig. 1(a). However, this will make data collecting challenging, even for unsupervised learning. As a result, many approaches based on few-shot learning at the price of accuracy have been proposed. For instance, Huang et al. (2022) employ meta-learning, as shown in Fig. 1(b). While due to complicated settings, it is impossible to migrate to the new product during the changeover flexibly, and the detection accuracy cannot be guaranteed. (2) In terms of training speed, when a large amount of data is utilized for training, the training progress for new goods is slowed in the actual production line. As is well-known, vanilla unsupervised AD requires to collect a large amount of information. Even though meta-learning works in few-shot learning, as shown in Fig. 1(b), it is still necessary to train a massive portion of previously collected data.

---

[*]Contributed Equally, [†]Corresponding Authors.

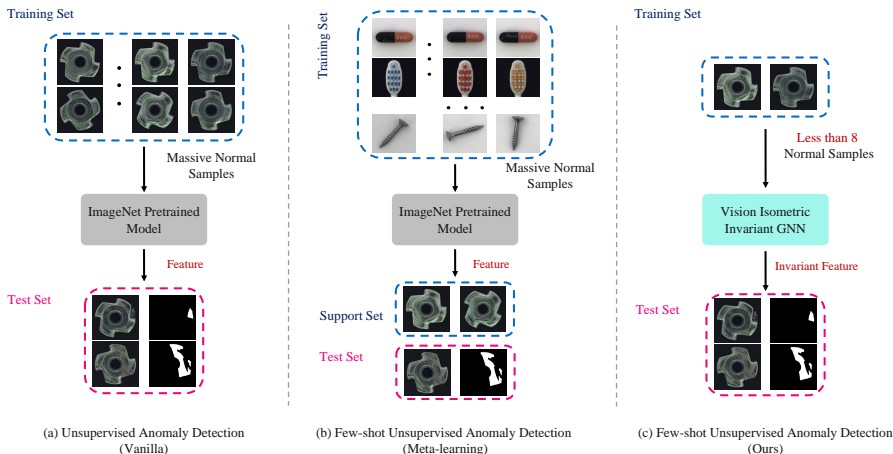

Figure 1: Different from (a) vanilla unsupervised AD and (b) few-shot unsupervised AD in meta learning. As input training samples, our setting (c) only utilizes a small number of normal samples. For our setting (c), there is no requirement to aggregate training categories in advance. The proposed model, vision isometric invariant GNN, can fast obtain the invariant feature within a few normal samples, and its accuracy outperforms models trained in a meta-learning context.

We state that AD of industrial products requires just a small quantity of data to achieve performance comparable to a large amount of data, i.e., a small quantity of image data can contain sufficient information to represent a large number of data. Due to the fact that industrial products are manufactured with high stability (no evident distortion of shape and color cast), the taken images lack the diversity of natural images, and there is a problem with the shooting angle or rotation. Therefore, it is essential to extract rotation-invariant structural features. As graph neural networks (GNNs) are capable of robustly extracting non-serialized structural features (Han et al. (2022), Bruna et al. (2013), Hamilton et al. (2017), Xu et al. (2018)), and they integrate global information better and faster Wang et al. (2020); Li et al. (2020). They are more suited than convolution neural networks (CNNs) to handle the problem of extracting rotation-invariant features. For this reason, the *core idea* of the proposed *GraphCore* method in this paper is to use the visual isometric invariant features (VIIFs) as the anomaly measurement features. In the method using memory bank ($\mathcal{M}$) as the AD paradigm, PatchCore (Roth et al. (2022)) uses ResNet (He et al. (2016)) as the feature extractor. However, since their features obtained by CNNs do not have rotation invariance (Dieleman et al. (2016)), a large number of redundant features are stored in $\mathcal{M}$. Note that these redundant features maybe come from multiple rotation features of the same patch structure. It will hence require a huge quantity of training data to ensure the high accuracy of the test set. To avoid these redundant features, we propose VIIFs, which not only produce more robust visual features but also dramatically lower the size of $\mathcal{M}$ and accelerate detection.

Based on the previous considerations, the goal of our work is to handle the cold start of the production line during the changeover. As shown in Fig. 1(c), a new FSAD method, called *GraphCore*, is developed that employs a small number of normal samples to accomplish fast training and competitive AD accuracy performance of the new product. On the one hand, by utilizing a small amount of data, we would rapidly train and accelerate the speed of anomaly inference. On the other hand, because we directly train new product samples, adaptation and migration of anomalies from the old product to the new product do not occur.

**Contributions.** In summary, the main contributions of this work are as follows:

- We present a feature-augmented method for FSAD in order to investigate the property of visual features generated by CNNs.

- We propose a novel anomaly detection model, GraphCore, to add a new VIIF into the memory bank-based AD paradigm, which can drastically reduce the quantity of redundant visual features.

- The experimental results show that the proposed VIIFs are effective and can significantly enhance the FSAD performance on MVTec AD and MPDD.

**Related Work.** Few-shot anomaly detection (FSAD) is an attractive research topic. However, there are only a few papers devoted to the industrial image FSAD. Some works (Liznerski et al. (2020); Pang et al. (2021); Ding et al. (2022)) experiment with few-shot abnormal images in the test set, which contradicts our assumptions that no abnormal images existed. While others (Wu et al. (2021); Huang et al. (2022)) conduct experiments in a meta-learning setting. This configuration has the disadvantage of requiring a high number of base class images and being incapable of addressing the shortage of data under cold-start conditions in industrial applications. PatchCore (Roth et al. (2022)), SPADE (Cohen & Hoshen (2020)), and PaDiM (Defard et al. (2021)) investigated AD performance on MVTec AD in a few-shot setting. However, these approaches are not intended for changeover-based few-shot settings. Thus their performance cannot satisfy the requirements of manufacturing changeover. In this research, we propose a feature augmentation method for FSAD that can rapidly finish the training of anomaly detection models with a small quantity of data and meet manufacturing changeover requirements.

## 2 APPROACH

**Problem Setting.** Fig. 1(c) outlines the formal definition of the problem setting for the proposed FSAD. Given a training set of only $n$ normal samples during training, where $n \leq 8$, from a specific category. At test time, given a normal or abnormal sample from a target category, the anomaly detection model should predict whether or not the image is anomalous and localize the anomaly region if the prediction result is anomalous.

**Challenges.** For the FSAD proposed in Fig. 1(c), we attempt to detect anomalies in the test sample using only a small number of normal images as the training dataset. The key challenges consist of: (1) Each category's training dataset contains only normal samples, i.e., no annotations at the image or pixel level. (2) There are few normal samples of the training set available. In our proposed setting, there are fewer than 8 training samples.

**Motivation.** In the realistic industrial image dataset (Bergmann et al. (2019); Jezek et al. (2021)), the images under certain categories are extremely similar. Most of them can be converted to one another with simple data augmentation, such as the meta nut (Fig. 2) and the screw (Fig. 6). For instance, rotation augmentation can effectively provide a new screw dataset. Consequently, when faced with the challenges stated in Section 2, our natural inclination is to acquire additional data through data augmentation. Then, the feature memory bank (Fig. 4) can store more useful features.

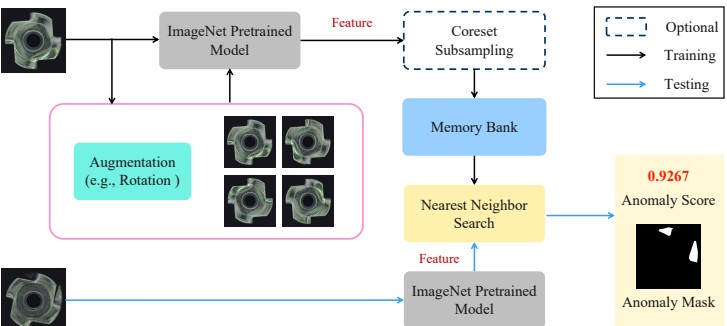

Figure 2: Augmentation+PatchCore Architecture.

## 2.1 AUGMENTATION+PATCHCORE

To validate our insight, we have adapted PatchCore (Roth et al. (2022)) to our model. We denote augmentation (rotation) with PatchCore as Aug.(R). The architecture is depicted in detail in Fig. 2. Before extracting features from the ImageNet pre-trained model, we augment the data (e.g., by rotating the data).

---

**Algorithm 1:** Aug.(R) memory bank

---

**Input** : ImageNet pre-trained $\phi$, all normal samples $\mathcal{X}_N$, data augmentation operator $\alpha$, patch feature extractor $\mathcal{P}$, memory size target $l$, random linear projection $\psi$.

**Output:** Patch-level augmented memory bank $\mathcal{M}$.

1 $\mathcal{M} \leftarrow \{\}$;
2 **for** $x_i \in \mathcal{X}_N$ **do**
3     $x_i^g \leftarrow \alpha(x_i)$;
4     $\mathcal{M} \leftarrow \mathcal{P}(\phi(x_i))$;
5     $\mathcal{M} \leftarrow \mathcal{P}(\phi(x_i^g))$;
6 **end for**
7 $\mathcal{M}_C \leftarrow \{\}$ //Apply coreset sampling for memory bank
8 **for** $i \in [0, \cdots, l-1]$ **do**
9     $m_i \leftarrow \underset{m \in \mathcal{M}-\mathcal{M}_C}{\arg\max} \ \underset{n \in \mathcal{M}_C}{\min} \ \|\psi(m) - \psi(n)\|_2$;
10     $\mathcal{M}_C \leftarrow \mathcal{M}_C \cup \{m_i\}$;
11 **end for**
12 $\mathcal{M} \leftarrow \mathcal{M}_C$.

---

In the training phase, the aim of the training phase is to build up a memory bank, which stores the neighborhood-aware features from all normal samples. At test time, the test image is predicted as anomalies if at least one patch is anomalous, and pixel-level anomaly segmentation is computed via the score of each patch feature. The feature memory construction method is shown in Algorithm 1. We default set ResNet18 (He et al. (2016)) as the feature extraction model. Conceptually, coreset sampling (Sener & Savarese (2018)) for memory bank aims to balance the size of the memory bank with the performance of anomaly detection. And the size of the memory bank has a considerable impact on the inference speed. In Section 3.3, we discuss the effect of the sampling rate in detail.

In testing phase, with the normal patches feature bank $\mathcal{M}$, the image-level anomaly score $s$ for the test image $x^{test}$ is computed by the maximum score $s^*$ between the test image's patch feature $\mathcal{P}(x^{test})$ and its respective nearest neighbour $m^*$ in $\mathcal{M}$.

From Table 2 and Table 3, we can easily observe that the performance of Aug.(R) greatly outperforms the SOTA models under the proposed few-shot setting.

## 2.2 VISION ISOMETRIC INVARIANT FEATURE

In Section 2.1, we heuristically demonstrate that Augmentation+PatchCore outperforms SOTA models in the few-shot anomaly detection context proposed. Essentially, the data augmentation method immediately incorporates the features of normal samples into the memory bank. In other words, Augmentation+PatchCore improves the probability of locating a subset feature, allowing the anomaly score of the test image to be calculated with greater precision. Therefore, we question whether it is possible to extract the invariant representational features from a small number of normal samples and add them to the feature memory bank. As demonstrated in Fig. 3, we propose a new model for feature extraction: vision isometric invariant graph neural network (VIIG). The proposed model is motivated by Section 2 and tries to extract the visual isometric invariant feature (VIIF) from each patch of the normal sample. As previously stated, the majority of industrial visual anomaly detection datasets are transformable via rotation, translation, and flipping. Thus, the isomorphism of GNN suited industrial visual anomaly detection excellently.

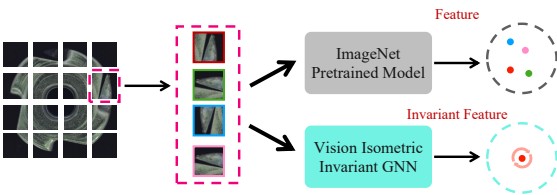

Figure 3: Convolution feature VS vision isometric invariant feature.

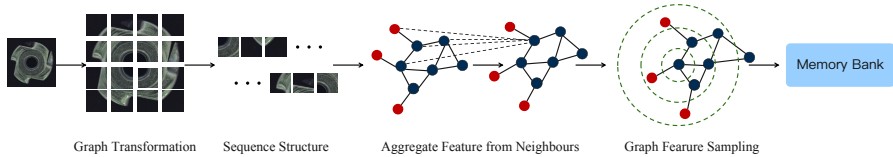

Figure 4: Vision isometric invariant GNN pipeline.

### 2.3 GRAPH REPRESENTATION OF IMAGE

Fig. 4 shows the feature extraction process of GraphCore. Specifically, for a normal sample image with a size of $H \times W \times 3$, we evenly separate it as an $N$ patch. In addition, each patch is transformed into a feature vector $f_i \in \mathbb{R}^D$. So we have the features $F = [f_1, f_2, \cdots, f_N]$, where $D$ is the feature dimension and $i = 1, 2, \cdots, N$. We view these features as unordered nodes $\mathcal{V} = \{v_1, v_2, \cdots, v_N\}$. For certain each node $v_i$, we denote the $K$ nearest neighbours $\mathcal{N}(v_i)$ and add an edge $e_{ij}$ directed from $v_j$ to $v_i$ for all $v_j \in \mathcal{N}(v_i)$. Hence, each patch of normal samples can be denoted as a graph $\mathcal{G} = (\mathcal{V}, \mathcal{E})$. $\mathcal{E}$ refers all the edges of Graph $\mathcal{G}$.

### 2.4 GRAPH FEATURE PROCESSING

Fig. 4 shows the architecture of the proposed vision isometric invariant GNN. To be specific, we set the feature extraction as GCN (Kipf & Welling (2017)). We aggregate features for each node by exchanging information with its neighbour nodes. In specific, the feature extraction operates as follows:

$$\mathcal{G}' = F(\mathcal{G}, \mathcal{W}) = Update(Aggregate(\mathcal{G}, W_{aggregate}), W_{update}), \tag{1}$$

where $W_{aggregate}$ and $W_{update}$ denote the weights of the aggregation and update operations. Both of them can be optimized in an end-to-end manner. Specifically, the aggregation operation for each node is calculated by aggregating neighbouring nodes' features:

$$f_i' = h(f_i, g(f_i, \mathcal{N}(f_i), W_{aggregate}), W_{update}), \tag{2}$$

where $h$ is the node feature update function and $g$ is the node feature aggregate feature function. $\mathcal{N}(f_i^l)$ denotes the set of neighbor nodes of $f_i^l$ at the $l$-th layer. Specifically, we employ max-relative graph convolution (Li et al. (2019)) as our operator. So $g$ and $h$ are defined as:

$$g(\cdot) = f_i'' = max(\{f_i - f_j | j \in \mathcal{N}(x_i)\}), \tag{3}$$

$$h(\cdot) = f_i' = f_i'' W_{update}. \tag{4}$$

In Equations 3 and 4, $g(\cdot)$ is a max-pooling vertex feature aggregator that aggregates the difference in features between node $v_i$ and all of its neighbours. $h(\cdot)$ is an MLP layer with batch normalization and ReLU activation.

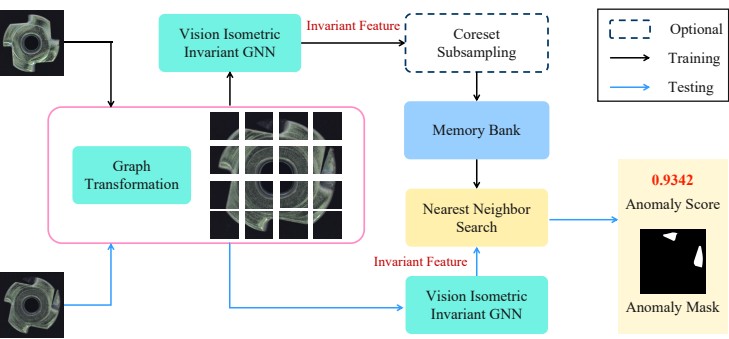

Figure 5: Vision isometric invariant GNN for FSAD.

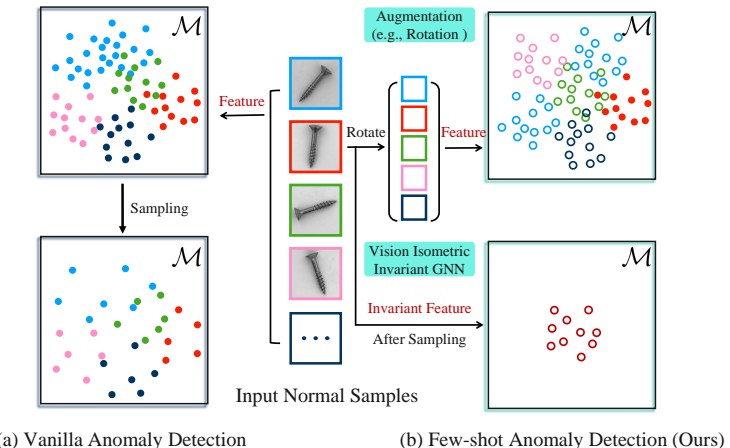

(a) Vanilla Anomaly Detection        (b) Few-shot Anomaly Detection (Ours)

Figure 6: Vision GNN architecture. (a) vanilla AD and (b) our proposed FSAD.

## 2.5 GRAPHCORE ARCHITECTURE

Fig. 5 shows the whole architecture of GraphCore. In the training phase, the most significant difference between GraphCore and Augmentation+PatchCore is the feature memory bank construction algorithm. The feature construction algorithm is the same as Aug.(R) memory bank in Algorithm 1. Note that we use vision isometric invariant GNN as feature extractor $\mathcal{P}$ without data augmentation. In the testing phase, the computation of anomaly score $s*$ for GraphCore is highly similar to the one in Augmentation+PatchCore. The only difference is the feature extraction method for each normal patch sample. The architecture details of the GraphCore are shown in the reference Table 21.

## 2.6 A UNIFIED VIEW OF AUGMENTATION+PATCHCORE AND GRAPHCORE

Fig. 6 depicts a unified view of both Augmentation+PatchCore and GraphCore. Augmentation+PatchCore prompts GraphCore to obtain the isometric invariant feature. Therefore, GraphCore can improve the probability of locating a feature subset, allowing the anomaly score of a test image to be calculated most precisely and rapidly. Table 1 shows the difference between PatchCore, Augmentation+PatchCore and GraphCore in terms of architectural details.

Table 1: Unified view for three methods.

| Augmentation | Network | Model |
|---|---|---|
| No | ImageNet Pre-trained Model | PatchCore |
| Rotation | ImageNet Pre-trained Model | Aug.(R) |
| No | GNN | GraphCore |

## 3 EXPERIMENT

### 3.1 EXPERIMENT SETTING

**Datasets.** To demonstrate the generalization of our proposed method, we conduct experiments on three datasets, namely MVTec AD (Bergmann et al. (2019)), MPDD (Jezek et al. (2021)) and MVTec LOCO AD (Bergmann et al. (2022)).

**Competing Methods.** RegAD (Huang et al. (2022)) is the SOTA FSAD method. It works under a meta-learning setting: aggregated training on multiple categories and adapting to unseen categories, using few-shot unseen images as a support set. However, our proposed few-shot setting utilizes only a few images as a training set and not several categories. Taking into account the fairness of the experiments, we reimplement the classical and SOTA approaches in the field of unsupervised anomaly detection, such as SPADE (Cohen & Hoshen (2020)), STPM (Wang et al. (2021a)), RD4AD (Deng

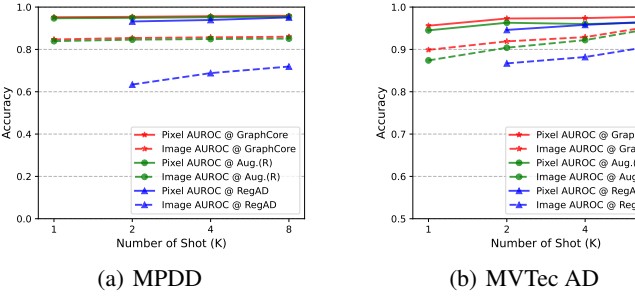

|           (a) MPDD            |           (b) MVTec AD           |

Figure 7: GraphCore VS Augmentation+PatchCore VS RegAD on various numbers of shot (K).

& Li (2022)), CFA (Lee et al. (2022)), and PatchCore (Roth et al. (2022)), using the official source code for comparison under our few-shot setting. PatchCore-1 is the result of our reimplementation with a 1% sampling rate, PatchCore-10 and PatchCore-25 are the results at 10% and 25% sampling rates, respectively, and RegAD-L is the RegAD experiment with our few-shot setting.

## 3.2 COMPARISON WITH THE SOTA METHODS

The comparative findings between MVTec and MPDD are shown in Table 2. Especially the performance of RegAD under the meta-learning setting is also listed in the table. In comparison to SOTA models, GraphCore improves average AUC by 5.8%, 4.1%, 3.4%, and 1.6% on MVTec and by 25.5%, 22.0%, 16.9%, and 14.1% on MPDD for 1, 2, 4, and 8-shot cases, respectively. From Fig. 7, it can be easily observed that GraphCore significantly outperforms the SOTA approach at the image and pixel level from 1-shot to 8-shot. As can be seen, the performance of GraphCore and Augmentation+PatchCore surpasses the other methods when using only a few samples for training.

Table 2: FSAD average results for all categories on MVTec AD and MPDD. The sampling ratio is 0.01, x|y represents image AUROC and pixel AUROC. The results for PaDiM, PatchCore-10 and PatchCore-25 are reported from Roth et al. (2022). The results for RegAD-L and RegAD are reported from Huang et al. (2022). The best-performing method is in bold.

| Dateset | K | Aug.(R) | GraphCore | CFA | SPADE | PaDiM | STPM | RD4AD | PatchCore-1 | PatchCore-10 | PatchCore-25 | RegAD-L | RegAD |
|---------|---|---------|-----------|-----|-------|-------|------|-------|-------------|--------------|--------------|---------|-------|
| MVTec AD | 1 | 87.4\|94.5 | **89.9\|95.6** | 78.8\|90.7 | 69.8\|79.1 | 76.1\|88.2 | 69.7\|58.2 | 74.4\|69.0 | 78.5\|90.1 | 83.4\|92.0 | 84.1\|92.4 | - | 82.4\|- |
|  | 2 | 90.4\|96.3 | **91.9\|96.9** | 81.1\|91.0 | 70.7\|79.9 | 78.9\|90.5 | 74.2\|59.8 | 75.5\|71.8 | 87.8\|94.8 | 86.4\|93.1 | 87.2\|93.3 | 81.5\|- | 85.7\|94.6 |
|  | 4 | 92.2\|96.0 | **92.9\|97.4** | 85.0\|91.3 | - | 71.6\|80.2 | 74.8\|60.8 | 76.9\|72.2 | 89.5\|95.0 | - | - | 84.9\|- | 88.2\|95.8 |
|  | 8 | 95.4\|96.4 | **95.9\|97.8** | 90.9\|91.6 | - | 75.3\|80.5 | 77.6\|61.6 | 78.5\|73.0 | 94.3\|95.6 | - | - | 87.4\|- | 91.2\|96.7 |
| MPDD | 1 | 83.9\|94.7 | **84.7\|95.2** | 58.8\|77.7 | - | 57.5\|73.9 | 59.2\|75.1 | 58.5\|73.2 | 59.2\|78.5 | - | - | - | 57.8\|- |
|  | 2 | 84.6\|94.9 | **85.4\|95.4** | 58.6\|78.2 | - | 58.0\|75.4 | 62.4\|75.8 | 61.8\|74.5 | 59.6\|79.2 | - | - | 50.8\|- | 63.4\|93.2 |
|  | 4 | 84.9\|95.2 | **85.7\|95.7** | 59.3\|78.7 | - | 58.3\|75.9 | 62.6\|76.2 | 62.1\|75.5 | 59.8\|79.8 | - | - | 54.2\|- | 68.8\|93.9 |
|  | 8 | 85.1\|95.5 | **86.0\|95.9** | 60.9\|79.0 | - | 58.5\|76.2 | 63.1\|76.6 | 62.4\|75.7 | 60.0\|80.3 | - | - | 61.1\|- | 71.9\|95.1 |

Considering that RegAD only shows detailed results of various categories above 2-shot, we only show the detailed results of 2-shot in the main text, and the results of 1-shot, 4-shot, and 8-shot are in the appendix. As shown in Table 3, GraphCore outperforms all other baseline methods in 12 out of the 15 categories at the image level and outperforms all other baselines in 11 out of the 15 categories at the pixel level on MVTec AD. Moreover, results in Table 4 show that GraphCore outperforms all other baselines in 5 out of the 6 categories at the image level and outperforms all other baselines in all categories at the pixel level on MPDD.

## 3.3 ABLATION STUDIES

**Sampling Rate.** When demonstrated in Fig. 8, our technique significantly improves as the sample rate increases from 0.0001 to 0.001, after which the increase in sampling rate has a flattening effect on the performance gain. In other words, as the sampling rate steadily increases, the performance of GraphCore is insensitive to the sampling rate.

**Nearest Neighbour.** In Fig. 8, the green colour represents the performance of GraphCore's 9 nearest neighbour search, and the blue colour represents the performance of GraphCore's 3 nearest neighbour search. As can be seen, increasing the number of neighbours from 3 to 9 greatly increases

Table 3: FSAD results on MVTec AD. The number of shots K is 2, and the sampling ratio is 0.01, x|y represents image AUROC and pixel AUROC. The results for PaDiM, PatchCore-10 and PatchCore-25 are reported from Roth et al. (2022). The results for RegAD are reported from Huang et al. (2022). The best-performing method is in bold.

| Category | Aug.(R) | GraphCore | CFA | SPADE | PaDiM | STPM | RD4AD | PatchCore-1 | PatchCore-10 | PatchCore-25 | RegAD |
|---|---|---|---|---|---|---|---|---|---|---|---|
| Bottle | 99.7\|98.6 | **99.8\|99.8** | 93.7\|93.5 | 95.7\|86.8 | - | 93.8\|84.6 | 91.2\|81.7 | 99.7\|98.5 | - | - | 99.4\|98.0 |
| Cable | 94.7\|96.2 | **95.2**\|96.3 | 89.3\|88.9 | 60.4\|78.6 | - | 60.2\|51.6 | 65.3\|65.4 | 94.9\|**97.8** | - | - | 65.1\|91.7 |
| Capsule | 66.5\|97.7 | **73.2\|97.8** | 53.4\|85.9 | 48.7\|79.8 | - | 45.2\|59.2 | 50.5\|78.2 | 67.2\|97.7 | - | - | 67.5\|97.3 |
| Carpet | **99.4**\|99.1 | **99.4\|99.6** | 97.6\|97.9 | 92.1\|95.6 | - | 90.8\|60.5 | 92.8\|74.2 | 99.1\|99.0 | - | - | 96.5\|98.9 |
| Grid | 75.7\|79.8 | 81.5\|80.6 | 80.4\|**81.4** | 75.8\|75.9 | - | 72.6\|61.2 | 75.2\|76.3 | 61.7\|67.5 | - | - | **84.0**\|77.4 |
| Hazelnut | **99.7**\|97.9 | 99.5\|**98.2** | 99.4\|**98.2** | 95.2\|88.9 | - | 90.3\|74.5 | 93.4\|64.8 | 93.5\|96.4 | - | - | 96.0\|98.1 |
| Leather | **100**\|99.3 | **100\|99.4** | **100**\|99.3 | 97.9\|89.2 | - | 95.8\|75.2 | 96.7\|86.5 | **100**\|99.3 | - | - | 99.4\|98.0 |
| Meta Nut | 95.0\|96.8 | **96.3\|98.1** | 68.6\|89.7 | 61.3\|59.5 | - | 59.4\|51.1 | 63.4\|68.9 | 92.0\|97.1 | - | - | 91.4\|96.9 |
| Pill | 87.8\|93.9 | **88.6**\|94.1 | 67.4\|91.5 | 60.2\|57.2 | - | 58.7\|49.9 | 62.8\|70.2 | 87.4\|**96.8** | - | - | 81.3\|93.6 |
| Screw | 63.6\|96.0 | **65.7**\|96.5 | 58.2\|**96.7** | 51.3\|70.2 | - | 51.9\|51.8 | 54.3\|60.8 | 48.3\|90.8 | - | - | 52.5\|94.4 |
| Tile | **100**\|99.3 | **100\|96.8** | 99.8\|81.8 | 90.2\|82.3 | - | 91.4\|58.2 | 88.9\|59.2 | **100**\|96.0 | - | - | 94.3\|94.3 |
| Toothbrush | 83.6\|98.2 | **87.3\|98.6** | 86.9\|93.9 | 80.2\|76.8 | - | 76.5\|66.3 | 77.1\|78.3 | 83.9\|98.2 | - | - | 86.6\|98.2 |
| Transistor | 96.3\|94.1 | **97.1\|99.2** | 72.5\|80.3 | 51.6\|73.6 | - | 82.4\|47.5 | 78.1\|67.8 | 96.9\|95.0 | - | - | 86.0\|93.4 |
| Wood | 97.1\|98.4 | 97.5\|**99.5** | 98.2\|92.4 | 50.3\|89.7 | - | 95.8\|48.4 | 93.7\|93.8 | 97.2\|93.0 | - | - | **99.2**\|93.5 |
| Zipper | 96.9\|99.0 | 97.5\|**99.3** | 50.5\|94.1 | 49.4\|93.7 | - | 47.6\|56.3 | 49.5\|51.2 | 95.3\|98.2 | - | - | 86.3\|95.1 |
| Average | 90.4\|96.3 | **91.9\|96.9** | 81.1\|91.0 | 70.7\|79.9 | 78.9\|90.5 | 74.2\|59.8 | 75.5\|71.8 | 87.8\|94.8 | 86.4\|93.1 | 87.2\|93.3 | 85.7\|94.6 |

Table 4: FSAD results on MPDD. The number of shots K is 2, and the sampling ratio is 0.01, x|y represents image AUROC and pixel AUROC. The results for PaDiM, PatchCore-10 and PatchCore-25 are reported from Roth et al. (2022). The results for RegAD are reported from Huang et al. (2022). The best-performing method is in bold.

| Category | Aug.(R) | GraphCore | CFA | SPADE | STPM | RD4AD | PatchCore | RegAD |
|---|---|---|---|---|---|---|---|---|
| Bracket Black | 66.8\|92.1 | 67.0\|**92.5** | 54.3\|75.8 | 62.4\|72.8 | **94.5**\|75.1 | 91.7\|75.4 | 58.6\|78.9 | 63.3\|- |
| Bracket Brown | 76.1\|91.9 | **77.2\|92.6** | 66.8\|77.5 | 59.5\|71.9 | 62.3\|73.2 | 58.8\|73.4 | 70.7\|76.9 | 59.4\|- |
| Bracket White | 87.2\|97.1 | **89.4\|97.5** | 68.7\|70.8 | 67.2\|72.4 | 53.8\|64.2 | 55.6\|62.4 | 70.4\|68.1 | 55.6\|- |
| Connector | 98.6\|97.2 | **98.9\|97.7** | 58.5\|88.2 | 59.2\|82.8 | 51.6\|83.4 | 53.7\|82.3 | 59.2\|85.2 | 73.0\|- |
| Metal Plate | **99.9**\|98.4 | **99.9\|99.1** | 62.7\|84.3 | 64.2\|75.9 | 62.4\|83.2 | 65.2\|76.5 | 64.1\|86.3 | 61.7\|- |
| Tubes | 79.2\|92.6 | **79.8\|93.1** | 40.7\|72.8 | 35.6\|76.8 | 49.6\|75.6 | 45.9\|77.1 | 34.3\|79.5 | 67.1\|- |
| Average | 84.6\|94.9 | **85.4\|95.4** | 58.6\|78.2 | 58.0\|75.4 | 62.4\|75.8 | 61.8\|74.5 | 59.6\|79.2 | 63.4\|93.2 |

performance at the pixel level when the sampling rate is low, but does not enhance performance at the image level. As the sampling rate increases, the gain of the number of pixels' neighbours approaches zero.

**Augmentation Methods.** Fig. 9 demonstrates that the performance of PatchCore on MVTec AD and MPDD is relatively weak, but Aug.(R) demonstrates higher performance. It demonstrates heuristically that our enhancement to feature rotation is significantly effective. Moreover, GraphCore outperforms Aug.(R) by a large margin, confirming our assumption that GraphCore can extract the isometric invariant feature from industrial-based anomaly detection images.

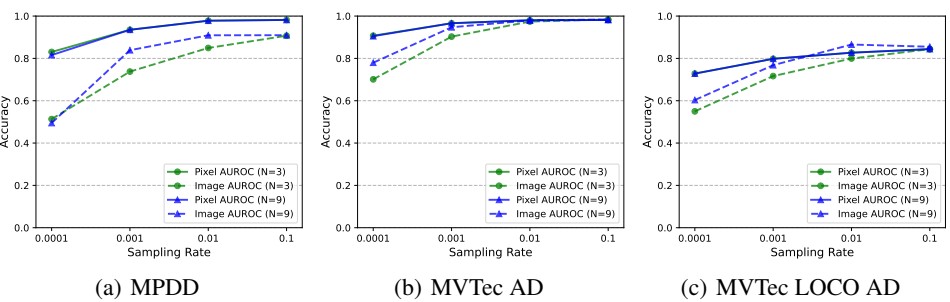

|  (a) MPDD | (b) MVTec AD | (c) MVTec LOCO AD |

Figure 8: Ablation results on sampling rates and the number of $N$ nearest neighbors.

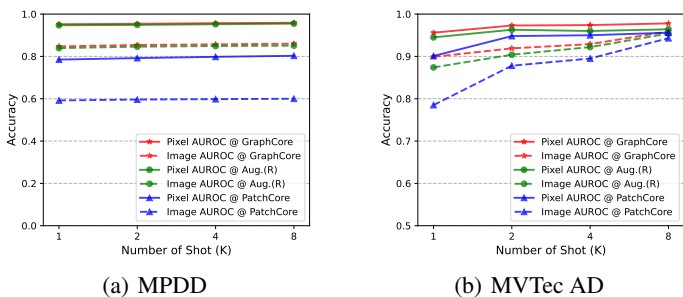

(a) MPDD

(b) MVTec AD

Figure 9: GraphCore vs Augmentation+PatchCore vs PatchCore on various number of shot (K).

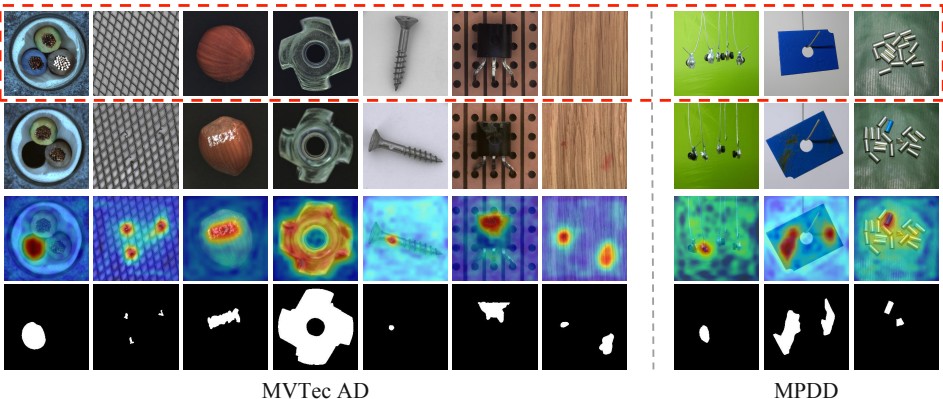

MVTec AD

MPDD

Figure 10: Visualization results of the proposed method on MVTec AD and MPDD. The first row denotes the training example in the 1-shot setting. The second row is test samples (abnormal). The third row is the heatmap on test samples. The fourth row is the anomaly mask (ground truth).

## 3.4 VISUALIZATION

Fig. 10 shows the visualization results obtained by our method on MVTec AD and MPDD with sampling rates of 0.01 and 1 shot, respectively. Each column represents a different item type, and the four rows, from top to bottom, are the detection image, anomaly score map, anomaly map on detection image, and ground truth. According to the results, our method can produce a satisfactory impact of anomaly localization on various objects, indicating that it has a strong generalized ability even in the 1-shot case.

## 4 CONCLUSION

In this study, we introduce a new approach, GraphCore, for industrial-based few-shot visual anomaly detection. Initially, by investigating the CNN-generated feature space, we present a simple pipeline - Augmentation+PatchCore - for obtaining rotation-invariant features. It turns out that this simple baseline can significantly improve anomaly detection performance. We further propose GraphCore to capture the isometric invariant features of normal industrial samples. It outperforms the SOTA models by a large margin using only a few normal samples ($\leq 8$) for training. The majority of industrial datasets for anomaly detection possess isomorphism, which is a property ideally suited to GraphCore. We will continue to push the limits of industrial-based few-shot anomaly detection in the future.

## 5 ACKNOWLEDGMENTS

This work is supported by the National Natural Science Foundation of China under Grant No. 62122035, 62206122, and 61972188. Y. Jin is supported by an Alexander von Humboldt Professorship for AI endowed by the German Federal Ministry of Education and Research.

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

# 6 APPENDIX

## 6.1 DATASET

**MVTec AD** is the most popular dataset for industrial image anomaly detection (Bergmann et al. (2019)), which consists of 15 categories of items, including a total of 3629 normal images as a training set, and a collection of 1725 normal images and abnormal images as a test set. All images have a resolution between 700×700 and 1024×1024 pixels.

**MPDD** is a more challenging AD dataset containing 6 classes of metal parts (Jezek et al. (2021)). The images are taken in different spatial directions and distances and under the condition of non-uniform background, so it is more challenging. The training set contains 888 normal images, and the test set contains 176 normal images and 282 abnormal images. The resolution of all images is 1024×1024 pixels.

**MVTEC LOCO AD** adds logical abnormal images outside the structural class abnormal image (Bergmann et al. (2022)). The dataset contains 1,772 normal images as a training set, and 304 normal images are used as a validation set. The test set contains 575 normal images, 432 structural abnormal images, and 561 logic abnormal images. Due to the different calculation methods of logic abnormal detection metric, we abandon the logical abnormal image of the test concentration, retaining the remaining 575 normal images and 432 structural abnormal images as a test set for experiments. Each image is 850 to 1600 pixels in height and 800 to 1700 pixels wide.

## 6.2 EXPERIMENT RESULTS

Table 5: Results of anomaly detection. Setting: New Few-shot Setting, K (number of shot)=1, Dataset: MVTec, Sampling Ratio: 0.01, Metrics: Image AUROC. The number of shot for RegAD is 2. The data for PaDiM and PatchCore-10, PatchCore-25 are from Roth et al. (2022).

| Category | Aug.(R) | GraphCore | CFA | SPADE | PaDiM | STPM | RD4AD | PatchCore-1 | PatchCore-10 | PatchCore-25 | RegAD |
|---|---|---|---|---|---|---|---|---|---|---|---|
| Bottle | 99.7 | **99.8** | 96.7 | 95.2 | - | 93.2 | 91.2 | 96.5 | - | - | - |
| Cable | 90.1 | **91.1** | 65.4 | 60.1 | - | 59.8 | 58.3 | 65.5 | - | - | - |
| Capsule | 64.7 | **72.1** | 50.2 | 45.6 | - | 43.2 | 44.7 | 49.8 | - | - | - |
| Carpet | **99.3** | **99.3** | 97.1 | 93.2 | - | 90.5 | 92.5 | 97.2 | - | - | - |
| Grid | 70.8 | **80.9** | 79.2 | 75.1 | - | 71.2 | 74.3 | 78.9 | - | - | - |
| Hazelnut | 97.4 | **98.5** | 98.1 | 95.0 | - | 90.3 | 93.2 | 98.0 | - | - | - |
| Leather | **100** | **100** | **100** | 97.2 | - | 95.1 | 96.5 | **100** | - | - | - |
| Meta Nut | 77.0 | **92.5** | 66.1 | 60.2 | - | 58.2 | 63.4 | 65.6 | - | - | - |
| Pill | 81.0 | 81.2 | 66.3 | 59.7 | - | 57.3 | 62.4 | 65.1 | - | - | - |
| Screw | 57.4 | **57.9** | 55.9 | 49.6 | - | 51.2 | 53.5 | 54.8 | - | - | - |
| Tile | **99.9** | 99.2 | 99.8 | 89.5 | - | 90.2 | 88.7 | 99.5 | - | - | - |
| Toothbrush | 84.4 | 85.2 | **86.7** | 78.5 | - | 75.2 | 77.8 | 85.8 | - | - | - |
| Transistor | 94.5 | **96.2** | 71.5 | 50.5 | - | 83.2 | 77.5 | 70.5 | - | - | - |
| Wood | 97.0 | 97.3 | 98.1 | 49.5 | - | 95.4 | 93.5 | 98.9 | - | - | - |
| Zipper | 97.4 | **97.5** | 50.3 | 48.5 | - | 45.2 | 48.6 | 51.2 | - | - | - |
| Average | 87.4 | **89.9** | 78.75 | 69.83 | 76.1 | 69.70 | 74.4 | 78.5 | 83.40 | 84.10 | 82.4 |

Table 6: Setting: Ours Few-shot Setting, K (number of shot)=1, Dataset: MVTec, Sampling Ratio: 0.01, Metrics: Pixel AUROC. The number of shot for RegAD is 2. The data for PaDiM and PatchCore-10, PatchCore-25 are from Roth et al. (2022).

| Category | Aug.(R) | GraphCore | CFA | SPADE | PaDiM | STPM | RD4AD | PatchCore-1 | PatchCore-10 | PatchCore-25 | RegAD |
|---|---|---|---|---|---|---|---|---|---|---|---|
| Bottle | 98.5 | **99.8** | 93.2 | 85.2 | - | 84.3 | 81.7 | 93.0 | - | - | - |
| Cable | 95.1 | **96.2** | 88.2 | 78.2 | - | 50.9 | 64.8 | 87.0 | - | - | - |
| Capsule | 97.7 | **98.1** | 85.6 | 79.2 | - | 49.2 | 77.9 | 87.7 | - | - | - |
| Carpet | 99.1 | **99.3** | 97.5 | 95.2 | - | 60.5 | 72.5 | 98.8 | - | - | - |
| Grid | 70.5 | 76.9 | 81.3 | 75.6 | - | 61.2 | 74.3 | **84.1** | - | - | - |
| Hazelnut | 97.1 | **98.5** | 98.1 | 88.2 | - | 73.3 | 63.2 | 97.5 | - | - | - |
| Leather | 99.3 | **99.5** | 99.2 | 88.3 | - | 75.1 | 86.5 | 99.2 | - | - | - |
| Meta Nut | 93.2 | 92.5 | 89.5 | 58.5 | - | 51.1 | 68.7 | 90.1 | - | - | - |
| Pill | 95.7 | **96.2** | 91.2 | 54.2 | - | 49.3 | 65.6 | 90.4 | - | - | - |
| Screw | 92.0 | 93.4 | 96.5 | 69.6 | - | 51.2 | 59.7 | **95.8** | - | - | - |
| Tile | 95.8 | **96.8** | 81.5 | 81.5 | - | 57.2 | 88.7 | 82.7 | - | - | - |
| Toothbrush | 97.9 | **98.5** | 93.8 | 75.5 | - | 65.2 | 77.8 | 93.0 | - | - | - |
| Transistor | 93.6 | **96.2** | 79.5 | 73.5 | - | 43.2 | 77.5 | 78.8 | - | - | - |
| Wood | 93.1 | **94.3** | 91.8 | 89.5 | - | 45.4 | 93.5 | 90.7 | - | - | - |
| Zipper | **98.5** | 97.5 | 93.2 | 93.5 | - | 55.2 | 48.6 | 94.0 | - | - | - |
| Average | 94.47 | **95.60** | 90.67 | 79.07 | 88.20 | 58.15 | 69.03 | 90.85 | 92.00 | 92.40 | - |

Table 7: Setting: Ours Few-shot Setting, K (number of shot)=2, Dataset: MVTec, Sampling Ratio: 0.01, Metrics: Image AUROC. The data for PaDiM and PatchCore-10, PatchCore-25 are from Roth et al. (2022).

| Category | Aug.(R) | GraphCore | CFA | SPADE | PaDiM | STPM | RD4AD | PatchCore-1 | PatchCore-10 | PatchCore-25 | RegAD |
|---|---|---|---|---|---|---|---|---|---|---|---|
| Bottle | 99.7 | **99.8** | 93.7 | 95.7 | - | 93.8 | 91.2 | 99.7 | - | - | 99.4 |
| Cable | 94.7 | **95.2** | 89.3 | 60.4 | - | 60.2 | 65.3 | 94.9 | - | - | 65.1 |
| Capsule | 66.5 | **73.2** | 53.4 | 48.7 | - | 45.2 | 50.5 | 67.2 | - | - | 67.5 |
| Carpet | **99.4** | **99.4** | 97.6 | 92.1 | - | 90.8 | 92.8 | 99.1 | - | - | 96.5 |
| Grid | 75.7 | 81.5 | 80.4 | 75.8 | - | 72.6 | 75.2 | 61.7 | - | - | **84.0** |
| Hazelnut | **99.7** | 99.5 | 99.4 | 95.2 | - | 90.3 | 93.4 | 93.5 | - | - | 96.0 |
| Leather | **100** | **100** | **100** | 97.9 | - | 95.8 | 96.7 | **100** | - | - | 99.4 |
| Meta Nut | 95.0 | **96.3** | 68.6 | 61.3 | - | 59.4 | 63.4 | 92.0 | - | - | 91.4 |
| Pill | 87.8 | **88.6** | 67.4 | 60.2 | - | 58.7 | 62.8 | 87.4 | - | - | 81.3 |
| Screw | 63.6 | **65.7** | 58.2 | 51.3 | - | 51.9 | 54.3 | 48.3 | - | - | 52.5 |
| Tile | **100** | **100** | 99.8 | 90.2 | - | 91.4 | 88.9 | **100** | - | - | 94.3 |
| Toothbrush | 83.6 | **87.3** | 86.9 | 80.2 | - | 76.5 | 77.1 | 83.9 | - | - | 86.6 |
| Transistor | 96.3 | **97.1** | 72.5 | 51.6 | - | 82.4 | 78.1 | 96.9 | - | - | 86.0 |
| Wood | 97.1 | 97.5 | 98.2 | 50.3 | - | 95.8 | 93.7 | 97.2 | - | - | **99.2** |
| Zipper | 96.9 | **97.5** | 50.5 | 49.4 | - | 47.6 | 49.5 | 95.3 | - | - | 86.3 |
| Average | 90.40 | **91.91** | 81.06 | 70.69 | 78.90 | 74.16 | 75.53 | 87.81 | 86.40 | 87.20 | 85.70 |

Table 8: Setting: Ours Few-shot Setting, K (number of shot)=2, Dataset: MVTec, Sampling Ratio: 0.01, Metrics: Pixel AUROC. The data for PaDiM and PatchCore-10, PatchCore-25 are from Roth et al. (2022).

| Category | Aug.(R) | GraphCore | CFA | SPADE | PaDiM | STPM | RD4AD | PatchCore-1 | PatchCore-10 | PatchCore-25 | RegAD |
|---|---|---|---|---|---|---|---|---|---|---|---|
| Bottle | 98.6 | **99.8** | 93.5 | 86.8 | - | 84.6 | 81.7 | 98.5 | - | - | 98.0 |
| Cable | 96.2 | 96.3 | 88.9 | 78.6 | - | 51.6 | 65.4 | **97.8** | - | - | 91.7 |
| Capsule | 97.7 | **97.8** | 85.9 | 79.8 | - | 59.2 | 78.2 | 97.7 | - | - | 97.3 |
| Carpet | 99.1 | **99.6** | 97.9 | 95.6 | - | 60.5 | 74.2 | 99.0 | - | - | 98.9 |
| Grid | 79.8 | 80.6 | **81.4** | 75.9 | - | 61.2 | 76.3 | 67.5 | - | - | 77.4 |
| Hazelnut | 97.9 | **98.2** | 98.2 | 88.9 | - | 74.5 | 64.8 | 96.4 | - | - | 98.1 |
| Leather | 99.3 | **99.4** | 99.3 | 89.2 | - | 75.2 | 86.5 | 99.3 | - | - | 98.0 |
| Meta Nut | 96.8 | **98.1** | 89.7 | 59.5 | - | 51.1 | 68.9 | 97.1 | - | - | 96.9 |
| Pill | 93.9 | 94.1 | 91.5 | 57.2 | - | 49.9 | 70.2 | **96.8** | - | - | 93.6 |
| Screw | 96.0 | 96.5 | **96.7** | 70.2 | - | 51.8 | 60.8 | 90.8 | - | - | 94.4 |
| Tile | 99.3 | 96.8 | 81.8 | 82.3 | - | 58.2 | 59.2 | 96.0 | - | - | 94.3 |
| Toothbrush | 98.2 | **98.6** | 93.9 | 76.8 | - | 66.3 | 78.3 | 98.2 | - | - | 98.2 |
| Transistor | 94.1 | **99.2** | 80.3 | 73.6 | - | 47.5 | 67.8 | 95.0 | - | - | 93.4 |
| Wood | 98.4 | **99.5** | 92.4 | 89.7 | - | 48.4 | 93.8 | 93.0 | - | - | 93.5 |
| Zipper | 99.0 | **99.3** | 94.1 | 93.7 | - | 56.3 | 51.2 | 98.2 | - | - | 95.1 |
| Average | 96.29 | **96.92** | 91.03 | 79.85 | 90.50 | 59.75 | 71.82 | 94.75 | 93.10 | 93.30 | 94.59 |

Table 9: Setting: New Few-shot Setting, K (number of shot)=4, Dataset: MVTec, Sampling Ratio: 0.01, Metrics: Image AUROC

| Category | Aug.(R) | GraphCore | CFA | SPADE | STPM | RD4AD | PatchCore | RegAD |
|---|---|---|---|---|---|---|---|---|
| Bottle | 99.7 | **99.8** | 94.2 | 95.8 | 93.9 | 92.1 | 99.6 | 99.4 |
| Cable | 94.1 | 95.2 | 91.2 | 61.3 | 61.3 | 68.4 | **97.4** | 76.1 |
| Capsule | 66.2 | **74.5** | 56.2 | 48.7 | 47.4 | 51.7 | 66.3 | 72.4 |
| Carpet | **99.6** | 99.4 | 97.6 | 92.5 | 91.5 | 93.2 | 99.0 | 97.9 |
| Grid | 77.9 | 81.6 | 81.5 | 76.2 | 75.3 | 76.4 | 63.0 | **91.2** |
| Hazelnut | **99.9** | 99.5 | 99.4 | 95.6 | 91.4 | 93.8 | 92.8 | 95.8 |
| Leather | **100** | **100** | **100** | 98.2 | 96.9 | 96.8 | **100** | **100** |
| Meta Nut | 95.9 | **96.2** | 91.3 | 62.5 | 60.8 | 65.3 | 94.7 | 94.6 |
| Pill | **89.3** | 88.2 | 85.6 | 61.8 | 61.3 | 62.8 | 89.0 | 80.8 |
| Screw | 63.9 | **68.9** | 49.2 | 52.9 | 52.8 | 55.7 | 54.1 | 56.6 |
| Tile | **100** | **100** | 99.8 | 91.3 | 90.4 | 90.8 | 100 | 95.5 |
| Toothbrush | 94.4 | **95.2** | 87.2 | 81.7 | 80.4 | 76.7 | **95.2** | 90.9 |
| Transistor | 98.5 | **99.2** | 95.8 | 52.5 | 82.4 | 79.3 | 98.4 | 85.2 |
| Wood | 97.4 | 97.9 | **98.6** | 51.4 | 95.8 | 94.2 | 97.4 | **98.6** |
| Zipper | 96.9 | **98.2** | 94.3 | 52.2 | 47.6 | 56.7 | 95.5 | 88.5 |
| Average | 92.22 | **92.92** | 84.97 | 71.64 | 74.77 | 76.93 | 89.49 | 88.23 |

Table 10: Setting: New Few-shot Setting, K (number of shot)=4, Dataset: MVTec, Sampling Ratio: 0.01, Metrics: Pixel AUROC

| Category | Aug.(R) | GraphCore | CFA | SPADE | STPM | RD4AD | PatchCore | RegAD |
|---|---|---|---|---|---|---|---|---|
| Bottle | 98.6 | **99.8** | 93.6 | 86.9 | 84.9 | 81.8 | 98.6 | 98.4 |
| Cable | 96.6 | 96.9 | 89.1 | 78.7 | 52.2 | 66.2 | **97.9** | 92.7 |
| Capsule | 97.7 | **97.9** | 86.2 | 80.1 | 59.3 | 78.4 | 97.7 | 97.6 |
| Carpet | 99.1 | **99.6** | 98.2 | 95.0 | 60.6 | 74.8 | 99.0 | 98.9 |
| Grid | 81.9 | 82.3 | 82.5 | 76.1 | 61.8 | 76.9 | 70.6 | **85.7** |
| Hazelnut | 98.3 | **99.1** | 98.5 | 89.1 | 74.9 | 65.2 | 97.0 | 98.0 |
| Leather | 99.3 | **99.6** | 99.3 | 89.3 | 75.3 | 86.7 | 96.9 | 99.1 |
| Meta Nut | 96.8 | **98.1** | 89.9 | 60.2 | 51.8 | 69.2 | 97.0 | 97.8 |
| Pill | 97.0 | **97.5** | 91.6 | 58.2 | 50.6 | 70.4 | 96.9 | 97.4 |
| Screw | 93.8 | **96.5** | 96.8 | 71.3 | 51.9 | 60.9 | 92.1 | 95.0 |
| Tile | 95.7 | **96.7** | 82.3 | 82.4 | 58.5 | 59.5 | 96.0 | 94.9 |
| Toothbrush | 98.8 | **98.9** | 94.2 | 76.9 | 66.9 | 78.9 | 98.8 | 98.5 |
| Transistor | 94.1 | **99.3** | 80.5 | 74.2 | 57.5 | 67.9 | 95.0 | 93.8 |
| Wood | 93.2 | **99.5** | 92.6 | 90.4 | 48.9 | 94.2 | 93.1 | 94.7 |
| Zipper | 98.4 | **99.3** | 94.8 | 93.8 | 56.4 | 52.3 | 98.3 | 94.0 |
| Average | 95.95 | **97.40** | 91.34 | 80.17 | 60.77 | 72.22 | 94.99 | 95.77 |

Table 11: Setting: New Few-shot Setting, K (number of shot)=8, Dataset: MVTec, Sampling Ratio: 0.01, Metrics: Image AUROC

| Category | Aug.(R) | GraphCore | CFA | SPADE | STPM | RD4AD | PatchCore | RegAD |
|---|---|---|---|---|---|---|---|---|
| Bottle | **100** | 99.8 | 95.1 | 95.9 | 94.1 | 92.8 | 99.6 | 99.8 |
| Cable | 94.1 | 95.2 | 91.8 | 63.5 | 62.6 | 69.2 | **97.4** | 80.6 |
| Capsule | 89.7 | **90.5** | 69.5 | 58.9 | 57.8 | 58.5 | 85.3 | 76.3 |
| Carpet | 98.5 | **99.5** | 97.6 | 92.7 | 91.6 | 93.8 | 99.0 | 98.5 |
| Grid | **92.7** | 92.3 | 85.6 | 77.3 | 76.9 | 77.9 | 83.1 | 91.5 |
| Hazelnut | **100** | **100** | 99.4 | 96.5 | 91.8 | 94.2 | 99.8 | 96.5 |
| Leather | **100** | **100** | **100** | 98.7 | 97.2 | 97.2 | **100** | **100** |
| Meta Nut | 96.8 | 97.9 | 92.3 | 68.9 | 61.3 | 65.6 | 95.1 | **98.3** |
| Pill | 90.1 | **91.1** | 88.9 | 63.9 | 64.2 | 63.6 | 89.6 | 80.6 |
| Screw | 79.4 | 80.1 | 65.4 | 56.4 | 55.9 | 59.3 | 74.1 | 63.4 |
| Tile | 99.3 | **100** | 99.8 | 91.8 | 91.2 | 91.2 | **100** | 97.4 |
| Toothbrush | 94.6 | 95.1 | 88.9 | 82.9 | 82.3 | 77.9 | 96.8 | **98.5** |
| Transistor | 98.2 | **99.2** | 96.2 | 58.9 | 84.6 | 81.2 | 98.9 | 93.4 |
| Wood | 98.7 | 98.9 | 98.9 | 61.3 | 95.8 | 95.6 | 97.5 | **99.4** |
| Zipper | 99.0 | **99.2** | 94.5 | 62.5 | 57.2 | 58.9 | 98.4 | 94.0 |
| Average | 95.41 | **95.92** | 90.93 | 75.34 | 77.63 | 78.46 | 94.31 | 91.21 |

Table 12: Setting: New Few-shot Setting, K (number of shot)=8, Dataset: MVTec, Sampling Ratio: 0.01, Metrics: Pixel AUROC

| Category | Aug.(R) | GraphCore | CFA | SPADE | STPM | RD4AD | PatchCore | RegAD |
|---|---|---|---|---|---|---|---|---|
| Bottle | 98.6 | **99.8** | 93.6 | 87.1 | 85.2 | 82.1 | 98.7 | 97.5 |
| Cable | 97.0 | 97.2 | 89.2 | 78.9 | 53.3 | 68.2 | **98.3** | 94.9 |
| Capsule | 98.3 | **98.5** | 86.5 | 80.2 | 59.3 | 78.5 | 98.4 | 98.2 |
| Carpet | 99.1 | **99.7** | 98.4 | 95.1 | 60.7 | 79.2 | 99.2 | 98.9 |
| Grid | 82.5 | 83.7 | 82.8 | 77.2 | 61.8 | 76.9 | 71.5 | **88.7** |
| Hazelnut | 98.4 | **99.2** | 98.6 | 89.5 | 74.9 | 65.5 | 97.2 | 98.5 |
| Leather | 99.4 | **99.6** | 99.4 | 90.2 | 75.3 | 86.9 | 99.4 | 98.9 |
| Meta Nut | 97.3 | **98.9** | 89.9 | 60.5 | 54.6 | 69.5 | 97.5 | 96.9 |
| Pill | 98.1 | **98.4** | 91.7 | 58.2 | 55.7 | 70.5 | 98.1 | 97.8 |
| Screw | 94.2 | 96.6 | 96.9 | 71.4 | 52.3 | 61.9 | 92.5 | **97.1** |
| Tile | 96.8 | **97.4** | 83.4 | 82.5 | 58.9 | 60.8 | 96.3 | 95.2 |
| Toothbrush | **99.2** | **99.2** | 94.5 | 77.2 | 66.9 | 79.1 | **99.2** | 98.7 |
| Transistor | 95.2 | **99.4** | 81.5 | 74.5 | 58.2 | 67.9 | 95.7 | 96.8 |
| Wood | 93.8 | **99.7** | 92.7 | 90.4 | 49.2 | 94.5 | 93.4 | 94.6 |
| Zipper | 98.6 | **99.7** | 94.9 | 94.2 | 57.8 | 52.8 | 98.6 | 97.4 |
| Average | 96.43 | **97.80** | 91.60 | 80.47 | 61.61 | 72.95 | 95.60 | 96.67 |

Table 13: Setting: New Few-shot Setting, K (number of shot)=1, Dataset: MPDD, Sampling Ratio: 0.01, Metrics: Image AUROC

| Category | Aug.(R) | GraphCore | CFA | SPADE | STPM | RD4AD | PatchCore | RegAD |
|---|---|---|---|---|---|---|---|---|
| Bracket Black | 64.8 | 65.9 | 64.1 | 62.1 | **93.2** | 91.2 | 58.2 | - |
| Bracket Brown | 75.0 | **76.8** | 65.4 | 59.2 | 59.8 | 58.3 | 70.6 | - |
| Bracket White | 88.6 | **89.2** | 68.2 | 68.2 | 43.2 | 44.7 | 69.3 | - |
| Connector | 98.3 | **98.7** | 58.5 | 58.5 | 90.5 | 92.5 | 59.0 | - |
| Metal Plate | **99.9** | **99.9** | 62.1 | 63.2 | 71.2 | 74.3 | 64.1 | - |
| Tubes | 76.6 | **77.8** | 34.2 | 33.8 | 65.1 | 44.2 | 34.1 | - |
| Average | 83.87 | **84.72** | 58.75 | 57.50 | 59.20 | 67.53 | 59.22 | 57.8 |

Table 14: Setting: New Few-shot Setting, K (number of shot)=1, Dataset: MPDD, Sampling Ratio: 0.01, Metrics: Pixel AUROC

| Category | Aug.(R) | GraphCore | CFA | SPADE | STPM | RD4AD | PatchCore | RegAD |
|---|---|---|---|---|---|---|---|---|
| Bracket Black | 91.7 | **92.3** | 75.2 | 72.4 | 74.5 | 72.5 | 78.8 | - |
| Bracket Brown | 91.8 | **92.2** | 77.2 | 71.8 | 72.9 | 72.3 | 76.8 | - |
| Bracket White | 97.0 | **97.3** | 69.8 | 65.4 | 63.1 | 61.3 | 67.8 | - |
| Connector | 97.0 | **97.5** | 88.9 | 82.4 | 82.1 | 81.7 | 85.0 | - |
| Metal Plate | 98.1 | **98.9** | 83.1 | 75.2 | 83.2 | 75.4 | 84.1 | - |
| Tubes | 92.4 | **92.8** | 71.7 | 76.2 | 74.5 | 76.1 | 78.2 | - |
| Average | 94.67 | **95.17** | 77.65 | 73.90 | 75.05 | 73.22 | 78.45 | - |

Table 15: Setting: New Few-shot Setting, K (number of shot)=2, Dataset: MPDD, Sampling Ratio: 0.01, Metrics: Image AUROC

| Category | Aug.(R) | GraphCore | CFA | SPADE | STPM | RD4AD | PatchCore | RegAD |
|---|---|---|---|---|---|---|---|---|
| Bracket Black | 66.8 | 67.0 | 54.3 | 62.4 | **94.5** | 91.7 | 58.6 | 63.3 |
| Bracket Brown | 76.1 | **77.2** | 66.8 | 59.5 | 62.3 | 58.8 | 70.7 | 59.4 |
| Bracket White | 87.2 | **89.4** | 68.7 | 67.2 | 53.8 | 55.6 | 70.4 | 55.6 |
| Connector | 98.6 | **98.9** | 58.5 | 59.2 | 51.6 | 53.7 | 59.2 | 73.0 |
| Metal Plate | **99.9** | **99.9** | 62.7 | 64.2 | 62.4 | 65.2 | 64.1 | 61.7 |
| Tubes | 79.2 | **79.8** | 40.7 | 35.6 | 49.6 | 45.9 | 34.3 | 67.1 |
| Average | 84.63 | **85.37** | 58.62 | 58.02 | 62.37 | 61.82 | 59.55 | 63.35 |

Table 16: Setting: New Few-shot Setting, K (number of shot)=2, Dataset: MPDD, Sampling Ratio: 0.01, Metrics: Pixel AUROC

| Category | Aug.(R) | GraphCore | CFA | SPADE | STPM | RD4AD | PatchCore | RegAD |
|---|---|---|---|---|---|---|---|---|
| Bracket Black | 92.1 | **92.5** | 75.8 | 72.8 | 75.1 | 75.4 | 78.9 | - |
| Bracket Brown | 91.9 | **92.6** | 77.5 | 71.9 | 73.2 | 73.4 | 76.9 | - |
| Bracket White | 97.1 | **97.5** | 70.8 | 72.4 | 64.2 | 62.4 | 68.1 | - |
| Connector | 97.2 | **97.7** | 88.2 | 82.8 | 83.4 | 82.3 | 85.2 | - |
| Metal Plate | 98.4 | **99.1** | 84.3 | 75.9 | 83.2 | 76.5 | 86.3 | - |
| Tubes | 92.6 | **93.1** | 72.8 | 76.8 | 75.6 | 77.1 | 79.5 | - |
| Average | 94.88 | **95.42** | 78.23 | 75.43 | 75.78 | 74.52 | 79.15 | 93.2 |

Table 17: Setting: New Few-shot Setting, K (number of shot)=4, Dataset: MPDD, Sampling Ratio: 0.01, Metrics: Image AUROC

| Category | Aug.(R) | GraphCore | CFA | SPADE | STPM | RD4AD | PatchCore | RegAD |
|---|---|---|---|---|---|---|---|---|
| Bracket Black | 66.9 | 67.8 | 54.9 | 62.4 | **94.5** | 91.9 | 58.9 | 63.8 |
| Bracket Brown | 76.5 | **77.8** | 66.8 | 59.5 | 62.4 | 59.0 | 70.8 | 66.1 |
| Bracket White | 87.5 | **89.6** | 71.1 | 67.5 | 54.2 | 55.7 | 70.7 | 59.3 |
| Connector | **98.9** | **98.9** | 58.7 | 59.5 | 52.1 | 54.4 | 59.4 | 77.2 |
| Metal Plate | **99.9** | **99.9** | 62.9 | 64.9 | 62.4 | 65.5 | 64.4 | 78.6 |
| Tubes | 79.6 | **80.0** | 41.1 | 35.9 | 50.2 | 46.2 | 34.5 | 67.5 |
| Average | 84.88 | **85.67** | 59.25 | 58.28 | 62.62 | 62.12 | 59.78 | 68.75 |

Table 18: Setting: New Few-shot Setting, K (number of shot)=4, Dataset: MPDD, Sampling Ratio: 0.01, Metrics: Pixel AUROC

| Category | Aug.(R) | GraphCore | CFA | SPADE | STPM | RD4AD | PatchCore | RegAD |
|---|---|---|---|---|---|---|---|---|
| Bracket Black | **92.7** | **92.7** | 75.9 | 72.9 | 75.3 | 75.9 | 79.1 | - |
| Bracket Brown | 92.1 | **92.9** | 77.9 | 72.3 | 73.5 | 74.8 | 77.3 | - |
| Bracket White | 97.5 | **97.8** | 71.2 | 72.9 | 64.7 | 64.5 | 69.3 | - |
| Connector | 97.5 | **98.1** | 88.8 | 82.9 | 84.2 | 82.4 | 86.4 | - |
| Metal Plate | 98.5 | **99.2** | 84.8 | 76.9 | 83.5 | 77.2 | 86.7 | - |
| Tubes | 92.7 | **93.5** | 73.5 | 77.2 | 75.8 | 78.1 | 80.1 | - |
| Average | 95.17 | **95.70** | 78.68 | 75.85 | 76.17 | 75.48 | 79.82 | 93.9 |

Table 19: Setting: New Few-shot Setting, K (number of shot)=8, Dataset: MPDD, Sampling Ratio: 0.01, Metrics: Image AUROC

| Category | Aug.(R) | GraphCore | CFA | SPADE | STPM | RD4AD | PatchCore | RegAD |
|---|---|---|---|---|---|---|---|---|
| Bracket Black | 67.1 | 68.2 | 55.2 | 62.4 | **94.6** | 92.2 | 59.2 | 67.3 |
| Bracket Brown | 76.8 | **78.5** | 66.9 | 59.8 | 62.7 | 59.2 | 70.9 | 69.5 |
| Bracket White | 87.9 | **89.9** | 79.4 | 67.6 | 54.7 | 55.9 | 70.5 | 61.4 |
| Connector | 98.9 | **99.1** | 58.9 | 59.9 | 52.7 | 54.6 | 59.6 | 84.9 |
| Metal Plate | **99.9** | **99.9** | 63.1 | 65.2 | 63.2 | 65.8 | 64.7 | 80.2 |
| Tubes | 79.8 | **80.3** | 41.7 | 36.2 | 50.8 | 46.7 | 34.8 | 67.9 |
| Average | 85.07 | **85.98** | 60.87 | 58.52 | 63.12 | 62.40 | 59.95 | 71.87 |

Table 20: Setting: New Few-shot Setting, K (number of shot)=8, Dataset: MPDD, Sampling Ratio: 0.01, Metrics: Pixel AUROC

| Category | Aug.(R) | GraphCore | CFA | SPADE | STPM | RD4AD | PatchCore | RegAD |
|---|---|---|---|---|---|---|---|---|
| Bracket Black | 92.9 | **92.9** | 76.2 | 73.1 | 76.3 | 76.2 | 79.6 | - |
| Bracket Brown | 92.3 | **93.1** | 77.9 | 72.6 | 74.2 | 75.1 | 77.5 | - |
| Bracket White | 97.9 | **98.2** | 71.8 | 73.1 | 64.9 | 64.6 | 70.2 | - |
| Connector | 98.1 | **98.3** | 89.1 | 83.1 | 84.5 | 82.6 | 87,1 | - |
| Metal Plate | 98.7 | **99.3** | 85.2 | 77.2 | 83.7 | 77.5 | 86.9 | - |
| Tubes | 92.9 | **93.4** | 73.9 | 78.1 | 76.1 | 78.3 | 80.5 | - |
| Average | 95.47 | **95.87** | 79.02 | 76.20 | 76.62 | 75.72 | 80.30 | 95.10 |

## 6.3 ARCHITECTURE DETAILS

Table 21: The architecture details of GraphCore

| Stage | Output Size | GraphCore |
|---|---|---|
| Stem | $\frac{H}{4} \times \frac{W}{4}$ | Conv $\times 3$ |
| Stage 1 | $\frac{H}{4} \times \frac{W}{4}$ | $\begin{bmatrix} D = 48 \\ K = 9 \end{bmatrix} \times 2$ |
| Downsample | $\frac{H}{8} \times \frac{W}{8}$ | Conv |
| Stage 2 | $\frac{H}{8} \times \frac{W}{8}$ | $\begin{bmatrix} D = 96 \\ K = 9 \end{bmatrix} \times 2$ |
| Downsample | $\frac{H}{16} \times \frac{W}{16}$ | Conv |
| Stage 3 | $\frac{H}{16} \times \frac{W}{16}$ | $\begin{bmatrix} D = 240 \\ K = 9 \end{bmatrix} \times 2$ |
| Downsample | $\frac{H}{32} \times \frac{W}{32}$ | Conv |
| Stage 4 | $\frac{H}{32} \times \frac{W}{32}$ | $\begin{bmatrix} D = 384 \\ K = 9 \end{bmatrix} \times 2$ |
| Head | $1 \times 1$ | Pooling and MLP |

In Table 21, D represents the feature dimension, whereas K represents the number of neighbors in GraphCore. $H \times W$ represents the size of the input image. We adapt GCN into the the pyramid architecture Wang et al. (2021b). The training epochs is 300. The optimizer is AdamW Loshchilov & Hutter. The batch size is 128. The initial learning rate is 0.005. The learning rate schedule is Cosine. The warmup epochs is 50. The weight decay is 0.05. The loss function is the cross entropy loss function.

## 6.4 ABLATION STADIES

Table 22: Ablation study for memory bank size and inference speed with respect to 1 shot

| Method | Memory Bank Size (Average) | Inference speed (Average) |
|---|---|---|
| PatchCore | 1.6M | 0.0316s |
| Aug.(R) + PatchCore | 1.8M | 0.0325s |
| GraphCore | 1.2M | 0.0299s |

Table 23: Ablation study for memory bank size and inference speed with respect to 2 shot

| Method | Memory Bank Size (Average) | Inference speed (Average) |
|---|---|---|
| PatchCore | 3.2M | 0.0327s |
| Aug.(R) + PatchCore | 3.2M | 0.0327s |
| GraphCore | 1.8M | 0.0287s |

The statical result of Table 22 and Table 23 clearly demonstrate the effectiveness of GraphCore, especially for memory bank size and its inference speed.

The statistical results presented in Tables 24 and 25 demonstrate that the rotation method outperforms the other augmentation techniques. We believe this indicates that the majority of industrial anomaly image datasets can be augmented by rotation. In the future, we believe that there will a more complex and realistic industrial anomaly image dataset that cannot be overcome by rotation.

Table 24: Ablation study with respect to Dataset: MVTec 2D, sampling rate: 0.01, Metrics: image-level AUROC, number of shot is 1.

| Augmentation Type | Aug + PatchCore |
|---|---|
| Flipping | 81.4 |
| Translation | 83.6 |
| Scaling | 82.3 |
| Rotation | 87.4 |

Table 25: Ablation study with respect to Dataset: MVTec 2D, sampling rate: 0.01, Metrics: image-level AUROC, number of shot is 2.

| Augmentation Type | Aug + PatchCore |
|---|---|
| Flipping | 83.7 |
| Translation | 85.6 |
| Scaling | 90.2 |
| Rotation | 90.5 |

