# OpenReview forum: "Pushing the Limits of Fewshot Anomaly Detection in Industry Vision: Graphcore"
_ICLR.cc/2023/Conference — ICLR 2023 poster_

### Official Review · Reviewer_Cvv9 · 2022-10-21

**Confidence:** 5
**Correctness:** 4
**Technical Novelty And Significance:** 3
**Empirical Novelty And Significance:** 3
**Recommendation:** 8

**Clarity, Quality, Novelty And Reproducibility:**

This paper is well-written and easy to understand overall. Regarding reproducibility, I think the author should provide source codes and give more visualizations of the training samples in the few-shot setting. Regarding novelty, the authors propose a novel backbone model to extract the feature for FSAD.


**Strength And Weaknesses:**

Strengths

+ The authors introduce one simple but effective method, Aug+PatchCore, which is well-suited for real-world challenges and provides valuable insight into the problems. The written logic is evident and easy to follow. I like the writing style of this paper and read it with great interest.
+ The authors propose an excellent problem setup to address fundamental challenges from a changeover-based setting. Previously, few-shot anomaly detection often adopts meta-learning settings, ignoring fast converagence in industrial manufacturing. Fig.1 is easy to follow. Moreover, the assumptions are reasonable.
+ The motivation is apparent and reasonable, where they find that the rotation-invariant features primarily increase the anomalies detection abilities. The authors utilize the isometric transformation property of GCN and propose a new feature extraction model, GraphCore. In addition, I believe that the introduction of GCN into anomaly detection for the first time is quite innovative.
+ The experiments are sufficient to support their conclusions.


Weaknesses

+ Though the numerical results are convincing, I think the authors had better visualize the normal samples in their training dataset, especially in the few-shot setting. I suppose that the quality of normal samples or features predominantly affects the result in one-shot or two-shot setting. So they need to show us which normal samples for training.
+  I encourage more data augmentation methods to sufficiently support their insights, not just for rotation augmentation. I think there is a need to explain why GraphCore could enhance the performance of other classes, like grid, bottle and even hazelnut. I suggest more data augmentation methods, e.g., flipping, translation or even scaling, should be conducted to prove the advantages of GraphCore. I think my suggestions could make this paper more substantial.

**Summary Of The Paper:**

This paper focuses on unsupervised visual anomaly detection in a changeover-based few-shot learning setting. It is a fundamental problem since the changeover scenario requires an anomaly detection model to localize the anomalies in a few samples. Unlike previous meta-learning settings, the authors propose a native few-shot setting, i.e., the training dataset at most contains eight normal samples, to simulate the changeover-based situation. The authors first introduce the data augmentation methods into PatchCore , which can significantly surpass the sota models, like RegAD. Then they unlock the deep insight from Aug+PatchCore and propose a novel model, GraphCore. The motivation of GraphCore is to utilize the rotation-invariant feature representation for each patch and replace ResNet with a GCN model. The authors evaluate their methods on multiple datasets and obtain a high performance of GraphCore in the changeover-based few-shot setting.

**Summary Of The Review:**

This is a vital paper, and I recommend it for acceptance. It addresses changeover-based few-shot anomaly detection problems in a practical manner. The authors propose a novel GraphCore model to extract visual isometric features. The extensive experiments verify the effectiveness of visual isometric features. However, the paper lacks the visualization of the training samples under one-shot and two-shot settings.  In total,  I would love to see GNN be more adapted to visual anomaly detection method, which boosts merging CV community with GNN community.

---

### Official Review · Reviewer_dzsB · 2022-10-21

**Confidence:** 4
**Correctness:** 4
**Technical Novelty And Significance:** 3
**Empirical Novelty And Significance:** 3
**Recommendation:** 8

**Clarity, Quality, Novelty And Reproducibility:**

The paper has a simple style and provides a clear description of the benefits of incorporating GNN into industrial visual AD. I believe that the method is novel. In addition, the paper covers the model's implementation and reproducibility.

**Strength And Weaknesses:**

Strength:
+ The scenario of the proposed few-shot AD is reasonable. It is concluded that using a small amount of training data is enough for achieving a satisfying performance.
+ The paper is the first one to introduce GNN into the field of industrial few-shot AD.
+ On the basis that feature rotation enhancement can bring gains to PatchCore, the paper designs a model named GraphCore to extract the most critical features of images without being affected by rotation. In this way, a small number of images can provide sufficient features for anomaly detection, reducing the size of memory bank.
+ The experiments are extensive. The paper compares the proposed method with a variety of representative methods on multiple datasets in terms with different shot numbers. It clearly proves the effectiveness of the paper's method.

Weaknesses:
+ There are relatively few GraphCore visualization results, and each category should have at least one visualization graph.
+ The structure of GNN is inadequately described. The author should provide additional details.
+ Grammatical errors. In the abstract, "provide a novel visual isometric invariant feature (VIIF) as anomaly measurement feature" should be "provide a novel visual isometric invariant feature (VIIF) as an anomaly measurement feature".

**Summary Of The Paper:**

This paper addresses the industrial visual few-shot anomaly detection (AD) in real-world challenges. In this situation, only a few normal samples are used for training. This paper first summarizes the previous methods and then point out why they are not suitable for the real-world scenario. The paper selects PatchCore as the baseline and analyzes the effect of feature enhancement on the memory bank. Then, the paper uses GNN from a different perspective to make image features unaffected by rotation, thereby reducing the dependence on feature enhancement methods. The paper designs a model named GraphCore for the changeover problem in industrial manufacturing, and obtains the new SOTA of few-shot AD.

**Summary Of The Review:**

The paper discusses anomaly detection challenges in the real world. The intent is plain. The comparison of the proposed strategy to previous approaches proves its effectiveness. This research, I believe, demonstrates the potential of GNN in the field of AD. However, the primary content need revisions to certain specifics.

---

### Official Review · Reviewer_oKEN · 2022-10-21

**Confidence:** 4
**Correctness:** 4
**Technical Novelty And Significance:** 3
**Empirical Novelty And Significance:** 3
**Recommendation:** 8

**Clarity, Quality, Novelty And Reproducibility:**

The clarity and quality of this paper are good. The basic idea is easy to follow. The novelty of this paper is outstanding. The rotation-invariant visual feature proposed by the author is desirable for solving the redundant feature problem. For reproducibility, the proposed method is solid and convincing, which can significantly improve anomaly detection performance in few-shot cases.


**Strength And Weaknesses:**

Strength:
+ Based on the actual industrial scene, the author designs a simple but efficient anomaly detection paradigm .
+ The author unlocks insights that the rotation-invariant visual features are critical for few-shot anomaly detection and then they propose a practical solution using graph neural networks.
+ I appreciate that the author clarifies the differences between convolution and isometric invariant features.
+ The experimental results show that the proposed model GraphCore has achieved significant performance with a considerable margin improvement.
+ Both ablation studies and visualization results verify the effectiveness of rotation-invariant features.
+ The paper is organized well and easy to understand.

Weaknesses:
- In Section 3.1, Dataset, the author lists three datasets. However, MVTec LOCO AD results are not reported in Section 3.2.
- The author states that visual features extracted by GraphCore is efficient. However, there are no ablation studies, such as the number of features used or feature visualization, to demonstrate this point.
- There are some typos as follows:
 (1) In Abstract, “we utilize of graph” should be “we utilize graph”.
 (2) In Section 2, Movtivation part, Fig. 5 has no the screw, and maybe Fig, 6?
 (3) In Figure 6, “Rotat” should be “Rotate”.

**Summary Of The Paper:**

In this paper, the author proposes a new setting for addressing industrial-based few-shot anomaly detection (FSAD). Specifically, the author leverages pre-trained graph neural networks to extract efficient features of normal products and anomalies, called visual isometric invariant features. Besides, the two proposed prototypes, i.e., convolution feature with augmentation and vision isometric invariant GNN (GraphCore), illustrate the significance of the proposed rotation-invariant feature in the AD task. Finally, ablation studies and experimental results are conducted comprehensively on popular datasets MVtecAD and MPDD.


**Summary Of The Review:**

Faced with the FSAD task in industrial manufacturing, the author proposes a simple setup and a solution called GraphCore that can capture rotation-invariant visual features. The effectiveness of the proposed method is validated through a comprehensive comparison of current FSAD methods, and the provided insights are well supported. This work is reasonable and solid overall.

---

### Official Review · Reviewer_Mmca · 2022-10-27

**Confidence:** 4
**Correctness:** 3
**Technical Novelty And Significance:** 3
**Empirical Novelty And Significance:** 3
**Recommendation:** 6

**Clarity, Quality, Novelty And Reproducibility:**

Clarity:
The paper is not clear about the description of the GCN for computing the VIIF.

Novelty:
The novelty is quite good and useful for improving the accuracy under the PatchCore framework.

Reproducibility:
It's difficult to reproduce the paper results, since there is lack of detailed description on the GCN architecture and how it is trained.



**Strength And Weaknesses:**

Strengths:
1. The idea of extracting visual isometric invariant features based on GNN architecture and applying it to the PatchCore framework for FSAD is novel and suitable for practical applications in industry.

2. The improvement of PatchCore with Aug.(R) memory bank is simple and useful to achieve better performance in accuracy and speed.

3. Experimental results on two public datasets by using the proposed methods show significant improvement over the previous SOTA methods for FSAD protocole.


Weaknesses:
1. The visual isometric invariant features (VIIF) is very critical in the proposed GraphCore method. Unfortunatley, the description on the VIIF is confusing. The equations (4)-(6) do not provide clear description on the GCN and some notations used in the equations are not well defined. In addition, there is no discussion on the training of the GCN model for computing the VIIF in the paper. What is the loss function for the network training?

2. The paper claims the proposed GraphCore method can provide fast model training, but there is no experimental justification on this claim.

3. How is the computational cost of the proposed methods compared to the competing SOTA methods, such as those different versions of PatchCore method? What are the sizes of the memory banks for the proposed methods and different versions of PatchCore method used in their experimental comparison.


**Summary Of The Paper:**

This paper presents a fewshot anomaly detection (FSAD) method based on the PatchCore framework. The main novelty in this paper is the proposed visual isometric invariant feature (VIIF) based on GNN  architecture in the proposed GraphCore method for few-shot anomaly detection. They show the proposed method can achieve SOTA results on two public datasets, including MVTec dataset.

**Summary Of The Review:**

In general, the proposed idea of using VIIF in the PatchCore framework is novel and useful. The experimental results show significant improvement over SOTA methods for FSAD experiments on public datasets. Unfortunately, the description of the GCN fopr VIIF is not clear. I am willing to raise my rating if more details can be provided for the GCN model in the paper.

---

### Decision · Program_Chairs · 2023-01-20

**Decision:**

Accept: poster

**Justification For Why Not Higher Score:**

- Contribution is specific to the industrial anomaly detection setting (which itself is a constrained case of general visual anomaly detection), which may not be of broad interest

**Justification For Why Not Lower Score:**

- Method is novel and well-motivated from the industrial anomaly detection setting
- Experiments show it is effective and outperforms competing methods


**Metareview: Summary, Strengths And Weaknesses:**

This paper proposes an algorithm for few-shot anomaly detection in the industrial context. Reviewers appreciated that the proposed method which uses graph neural networks was novel and well-motivated to the industrial anomaly detection setting, and that the experimental results show that it is effective. There were initial concerns that the description of the VIIF/GNN component were not clear and more analysis should be included but these were addressed in the authors' response to the reviewers' satisfaction. Overall, reviewers unanimously recommend acceptance. Authors are encouraged to incorporate these clarifications on the key VIIF/GNN component and proofread the final version of the paper.

**Note From Pc:**

if the above contains the word "oral" or "spotlight" please see: "oral" presentation means -> notable-top-5% and "spotlight" means -> notable-top-25%. As stated in our emails, we are disassociating presentation type from AC recommendations